# Analysis of Shielding Performance of Radiation-Shielding Materials According to Particle Size and Clustering Effects

Seon-Chil Kim

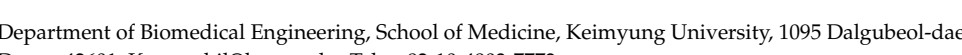

Department of Biomedical Engineering, School of Medicine, Keimyung University, 1095 Dalgubeol-daero, Daegu 42601, Korea; chil@kmu.ac.kr; Tel.: +82-10-4803-7773

**Abstract:** In the field of medical radiation shielding, there is an extensive body of research on process technologies for ecofriendly shielding materials that could replace lead. In particular, the particle size and arrangement of the shielding material when blended with a polymer material affect shielding performance. In this study, we observed how the particle size of the shielding material affects shielding performance. Performance and particle structure were observed for every shielding sheet, which were fabricated by mixing microparticles and nanoparticles with a polymer material using the same process. We observed that the smaller the particle size was, the higher both the clustering and shielding effects in the high-energy region. Thus, shielding performance can be improved. In the low-dose region, the effect of particle size on shielding performance was insignificant. Moreover, the shielding sheet in which nanoparticles and microsized particles were mixed showed similar performance to that of the shielding sheet containing only microsized particles. Findings indicate that, when fabricating a shielding sheet using a polymer material, the smaller the particles in the high-energy region are, the better the shielding performance is. However, in the low-energy region, the effect of the particles is insignificant.

**Keywords:** medical radiation; tungsten; clustering effect; shielding materials; shielding performance





## 1. Introduction

In the medical field, radiation is the most important tool for the diagnosis and treatment of diseases, and it is developing into a specialization. As the medical and industrial radiation fields are expanding, exposure to radiation from medical [1,2] and industrial [3,4] diagnostic devices is also increasing. Exposure to medical radiation occurs at low doses and only in some parts of the body. However, owing to the inadequate protection and carelessness of medical workers, the amount of exposure increases [5]. In addition, there is an increasing risk of cumulative radiation exposure due to the increasing frequency of examinations and photographs [6].

The medical radiation-exposure level is primarily determined by secondary scattered radiation, which is known as a low-dose region. Recently, attempts were made to reduce the exposure of medical workers to low-dose radiation while ensuring safety with a lightweight shielding suit [7–9]. Shielding clothing in medical institutions is made primarily of lead; however, because of issues such as the weight of lead, its toxic potential, manufacturing contaminations, and disposal costs, interest in developing a shield using ecofriendly materials is rising [10,11]. The most common ecofriendly radiation-shielding materials include tungsten, bismuth, barium sulfate, tin, and antimony [12,13].

Shields produced using these materials are primarily manufactured in the form of sheets, fibers, and thin films, and they have similar shielding performance to that of lead; thus, they are highly effective. Research into developing an ecofriendly shielding material to replace lead, and reduce the weight and thickness of clothing is underway [14]. This is important for expanding the range of users, ensuring that they can wear such clothing for a long time, and to maintain activity.

Research into increasing shield density to create lightweight and thin films is in progress; density depends on the mixing process and the process technology of the shielding material [15]. The shielding sheet, which is the most common apron material among medical radiation shields, is produced by either extrusion molding, calendering, coating, or injection molding by mixing a shielding material with a polymer resin. In addition, recent manufacturing practices include a laminated structure to absorb and control radiation, thus maintaining stable shielding performance [16].

Shielding sheets for medical radiation are primarily used as apron materials, and generally feature a shielding performance of 0.25 and 0.50 mmPb, lead-equivalent [17]. The thickness of the shielding sheet is around 3 mm, and is primarily determined by the amount of polymer and shielding materials. In previous studies, researchers investigated a method for reducing the thickness of the shielding sheet to increase density by reducing the particles of the shielding material rather than the type of polymer material [18–20].

To increase the density of the shielding sheet, process technology for lowering the porosity and increasing the particle packing is required [21]. To reduce the porosity, a method that narrows the gap between particles by reducing the particle size is used. However, most of the particles of commercial shielding sheets use microsized particles because they are more cost-effective, and these shielding sheets offer shielding performance that is similar to that of shielding sheets that use nanosized particles [22,23]. Therefore, this study evaluated shielding performance according to the particle size of the shielding material, and explains its quantitative effect in the process technology. In addition, this study attempts to confirm, through the cross-section of the shielding sheet, the clustering effect that narrows the gap between particles that occurs frequently when nanoparticles and microparticles are used. Thus, the effect of particle size on shielding performance is explained through testing the shielding effect. This study also demonstrates the mixing effect of the shielding-material particle size through experimentation, and improves the fabrication process for shielding sheets used in medical institutions.

## 2. Materials and Methods

When the shielding-material particles of the medical radiation shield are smaller, the distance between particles (DBP) decreases, and the incidental radiation energy is attenuated because of the collision between the photons and particles of the shielding material [24]. This is effective when small particles have a dense structure inside the shield. The size and arrangement of particles can actively react with radiation; therefore, better technology is required to increase shield durability. As shown in Figure 1, the interaction probability of incidental radiation on particles increases according to the arrangement and size of the particles [25].

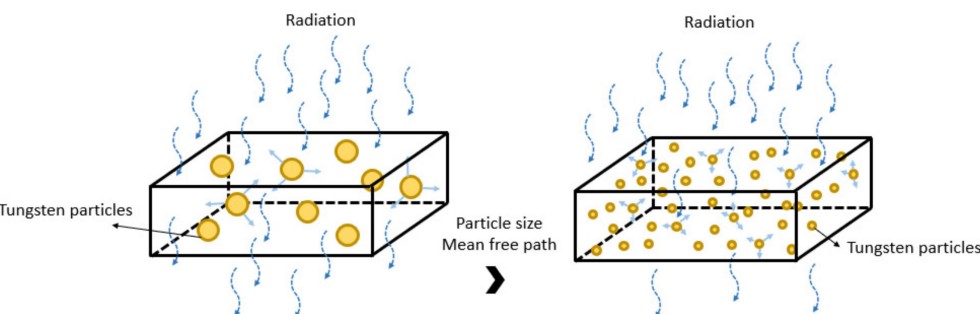

**Figure 1.** Radiation shielding according to particle size and arrangement of shielding material.

In this study, we assumed that uniformly isotropic shielding material particles exist within a certain space of the shielding sheet. At this time, the probability that the radiation particles reach distance $x$ without interaction is $P(x)$, and the probability of reaching the distance $x + dx$ is $P(x + dx)$. If the probability of interaction between the radiation particles

and the particles of the shielding material at a distance $dx$ is $\mu dx$, then the probability of not interacting is $1 - \mu dx$, and Equation (1) is established.

$$P(x + dx) = P(1 - \mu dx), \tag{1}$$

where $dx$ is the thickness of the shielding sheet. Therefore, if $P(0) = 1$, Equation (2) is established as follows:

$$P(x) = e^{-\mu x}. \tag{2}$$

Assuming that the radiation flux is a single energy, flux $f_0$ of unscattered particles passing through a certain thickness can be represented by $x = 0$, and the first flux by $f_0 \times c^{-\mu x}$.

$$f_0(x) = f_0(0)e^{-\mu x}, \tag{3}$$

where $\mu x$ represents the thickness of the shield; the greater the probability of collision is with shielding material particles per unit area when radiation passes inside the shield, the more shielding is achieved. Therefore, the distance when interacting with the shielding-material particles can be described as the mean free path of the medium; this mean free path is related to the thickness of the shield [26]. If the particles have an uneven arrangement, the interaction becomes a function of $x$ and is subject to Equation (4).

$$P(x) = exp\left[-\int_0^x \mu(x)dx\right]. \tag{4}$$

Lastly, if the interaction while passing through thickness $dx$ is regarded as the influence of the medium regardless of uniformity, it is expressed as Equation (5). This is a stochastic interaction depending on the thickness of the shield because the particles of the shielding materials are nonuniformly arranged per unit area of the shield. Since the nonuniformity of the medium corresponds to the number of particles in the unit area, the nonuniformity in the medium can be explained as shown in $\mu_i x_i$.

$$f_0(x) = f_0(0)exp\left[-\sum_i \mu_i x_i\right]. \tag{5}$$

Therefore, it is theoretically desirable to increase the number of particles, such as in Equation (5), to generate a large amount of interaction through the flux and attenuation media.

In this study, experiments were conducted to evaluate the multidispersed arrangement of particles according to the particle size of the shielding material, and the shielding performance according to the clustering effect. Tungsten was selected as the shielding material; tungsten has an atomic number of 74 and density of 19.25 g/cm$^3$, which makes it an effective alternative to lead [27,28]. First, as shown in Table 1, the purchased tungsten particles were processed into micro- and nanosizes of 100–400 μm and 400–900 nm, respectively, by ball-milling and laser-scattering methods (Microtrac Co., model S-3500) [29,30].

**Table 1.** Shielding-material particle-processing conditions.

|  | Microparticle | Nanoparticle |
|---|---|---|
| Particle size distribution (μm) | 100–400 | 0.4–0.9 |
| Specific surface area (m$^2$/g) | 24.45–31.2 | 8.5–10.5 |
| Tap density (g/cc) | 5.2–7.2 | 2.5–3.7 |
| Purity (%) | 99–99.8 | 99–99.8 |

To evaluate the shielding performance of a shielding sheet and quantitatively identify the particle distribution in the shielding sheet, we produced shielding sheets of three particle types (nanoparticles, microparticles, and a mix of nanoparticles and microparticles). We used high-density polyethylene (HDPE) as the polymer material for stirring with the shielding material during the manufacturing process; HDPE has excellent strength and is

used primarily as a disposable plastic product [31]. The HDPE used in this study had a molecular weight of 4 million or more and a density of 0.91 g/cm$^3$.

The material composed of tungsten particles was dried at 70 °C for 12 h prior to use. As a solid polymer was used, a casting solution was prepared using N-dimethylformamide (DMF, 99.5%) as a solvent. DMF was dissolved in a stirrer with polyethylene at a ratio of approximately 10 wt % to produce the casting solution. In addition, tungsten powder was added to the finished casting solution and stirred at 5000 rpm to disperse the tungsten particles. The plasticizer used to remove the porosity inside the shielding sheet was di-isononyl phthalate, and it was used in the range of 0.85–0.95 wt %. To maintain the uniform shielding performance of the final casting solution, we used a filter to remove foreign substances, and then performed a bubble removal operation. The final shielding sheet was finished with a calendar process of compression molding, and the size of the final shield sheet was 100 × 100 × 0.3 mm, which is the same as that shown in Figure 2.

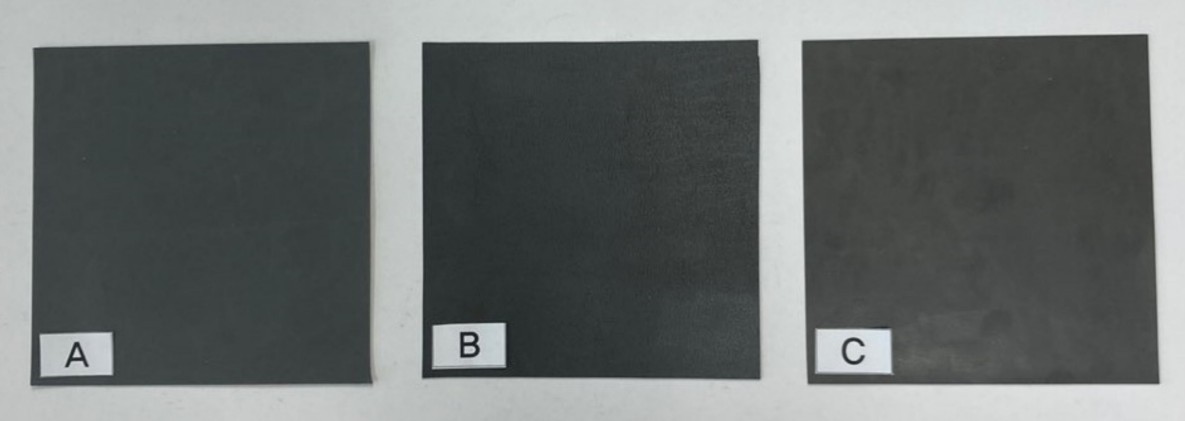

**Figure 2.** Appearance of final shielding sheet. (**A**) microparticle sheet; (**B**) nanoparticle sheet; (**C**) mixed nanoparticles and microparticles sheet).

In this experiment, changes in shielding efficiency according to particle size and arrangement were observed. In particular, the total amount of tungsten was adjusted to the same thickness of a single structure to estimate the appropriate particle size when manufacturing lightweight shielding clothing used in medical institutions. Three types of shielding sheets were manufactured, namely, a shielding sheet containing microsized particles, a sheet containing nanosized particles, and a sheet developed by mixing equal amounts of nanoparticles and microparticles. The three types of sheets had the same size and thickness, but the weights differed slightly depending on the mixing amount of the shielding material. The particle size and dispersibility of the shielding material were observed using a field-emission scanning electron microscope (FESEM; Hitachi, S-4800) through thin-film intercepts of the shielding sheet. Shielding efficiency was tested 10 times using an X-ray generator (Toshiba E7239, 150 kV-500 mA, 1999, Tokyo, Japan), and the average value was used. X-ray energy shielding test conditions were as follows: tube voltage, 60–120 kVp; tube current, 200 mA; and irradiation time, 0.1 s. The dose detector was DosiMax plus 1 (2019.iba Dosimetry Corp., Schwarzenbruck, Germany), and it was used after inspection and calibration. The radiation-shielding test was evaluated by applying the lead-equivalent test method (KS A 4025: confirmed in 1990, 1995) of X-ray protection products of the Korean industrial standard. In addition, the photon-scattering effect was measured at a distance of 50 cm, as shown in Figure 3, to prevent backscattering on the detector.

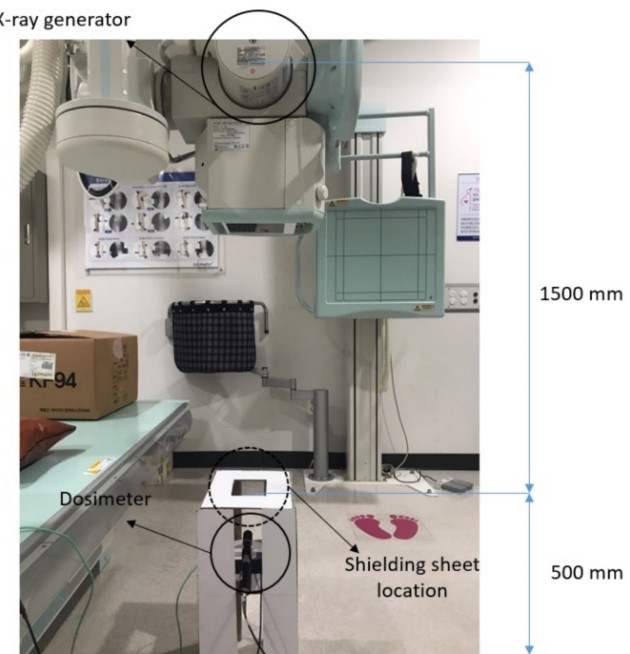

**Figure 3.** Evaluation of radiation-shielding performance of shielding sheets.

To evaluate shielding performance, the experiment consisted of a radiation experiment [32,33], as shown in Figure 3, and the shielding-efficiency measurement of the fabricated shielding sheet was calculated as shown in Equation (6).

$$S = \left(1 - \frac{e}{e_0}\right) \times 100, \tag{6}$$

where S is shielding efficiency, $e$ is the incident dose (mR), and $e_0$ is the transmitted dose (mR).

Here, $e$ is the exposure dose measured when there is a shielding sheet between X-ray beam and detector, and $e_0$ is the exposure dose measured when there is no shielding sheet between X-ray beam and detector [31]. Through this process, changes in particle size, array composition, and shielding efficiency in the shielding sheet were observed.

## 3. Results

The composition of the three types of shielding sheets manufactured using the same process is reported in Table 2. The appearance and mixing ratio of the sheets were the same, but there was a difference in the particle packing of tungsten and the weight per unit area. In general, a larger amount of shielding material can be added to a sheet composed of only nanoparticles.

**Table 2.** Fabrication composition of radiation-shielding sheet.

| Item | Value | | |
|---|---|---|---|
| | **A** | **B** | **C** |
| Sheet structure | | Single structure | |
| Shielding material | | Tungsten | |
| Mixing ratio (polymer: tungsten) | | 1:3 | |
| Sheet weight (kg/m$^2$) | 1.0 | 1.2 | 1.1 |
| Sheet thickness (mm) | 0.30 ± 0.005 | 0.32 ± 0.005 | 0.31 ± 0.005 |
| Solvents (g) | 11.2 | 12.1 | 11.4 |
| Polyethylene (g) | 23.4 | 22.5 | 25.1 |
| Tungsten (g) | 70.2 | 76.4 | 73.1 |

A, microparticle sheet; B, nanoparticle sheet; C, mixed microparticle and nanoparticle sheet.

Figure 4 shows the cross-sectional structure of the three shielding sheets with different tungsten particle contents, sizes, and arrangements. The comparison of the cross-sectional structures at the same magnification of the electron microscope shows that the array was composed of a dense structure when the particles were small. Figure 4C shows that it could not be arranged by particle size because it was artificially arranged, and that the dispersion of the shielding material according to particle size was not achieved during the stirring process.

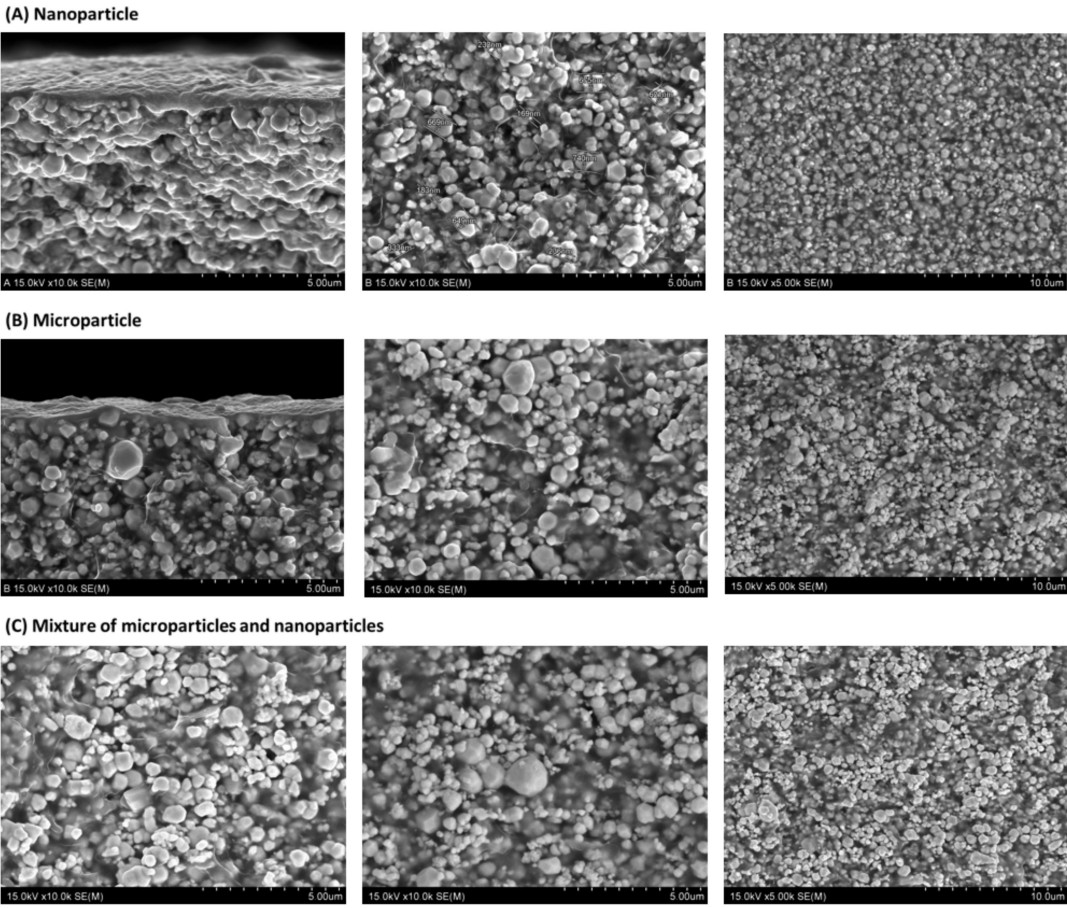

**Figure 4.** Structure of inner particle arrangement of shielding sheet. (**A**) Nanoparticle shielding sheet; (**B**) microparticle shielding sheet; (**C**) nano-micro particle mix sheet.

During sheet manufacturing, HDPE, the base material, was used to hold the particles of the shielding material. Figure 5 shows that the clustering effect between shielding material and polymer was more pronounced when nanoparticles were used. This shows that, when nanoparticles are used, it is ultimately effective in blocking pinholes through which radiation particles are transmitted. The clustering effect [34] between particles inside the shielding sheet explains the phenomenon in which particles are attached because the gap between particles is narrow, as shown in Figure 5A. Figure 5B shows that the gap between particles is narrow, but the phenomenon of particles sticking together is less than that of Figure 5A. Therefore, as confirmed by observations of the internal structure, if the size of the particles was small, a multilayered structure was formed in the sheet because of the clustering effect, and shielding efficiency improved. This is expected to increase the probability of particle interaction.

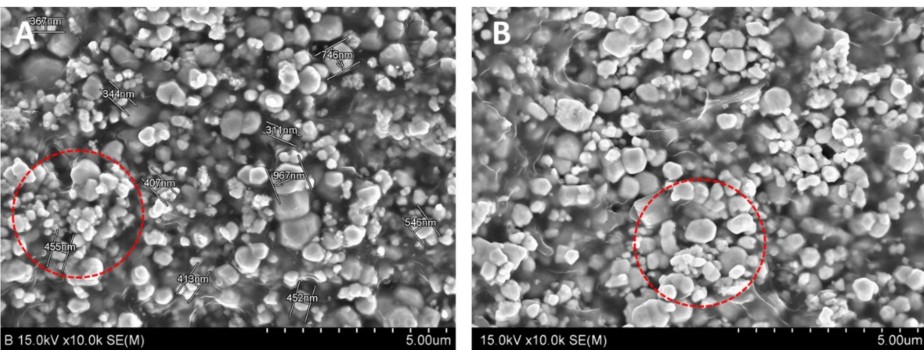

**Figure 5.** Comparison of nanoparticles and microparticles in cross-section of shielding sheet. (**A**) Nanoparticle shielding sheet; (**B**) microparticle shielding sheet (red circle is where clustering effect was observed).

Figure 6 shows the shielding-efficiency comparison of the three types of shielding sheets. Overall, the larger the particle size was, the greater the shielding efficiency; however, the higher the X-ray tube voltage was (i.e., the higher the energy intensity), the lower the shielding efficiency. With nanoparticles, the particles were small, and particle packing was high, making shielding efficiency high at high tube voltage. However, with microparticles, shielding efficiency tended to drop by approximately 5% at a high tube voltage. In the mixed nanoparticle and microparticle sheet, shielding efficiency was similar to that of the microparticle sheet.

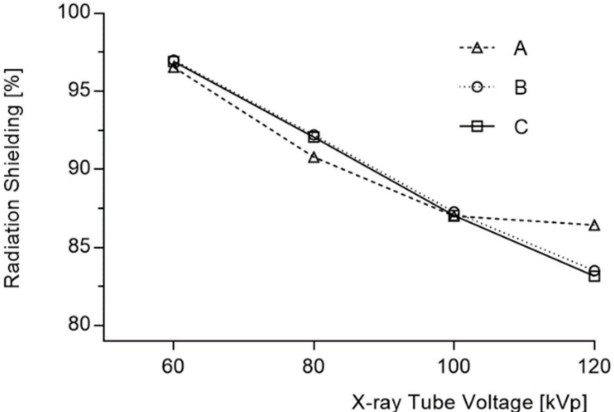

**Figure 6.** Change in shielding efficiency based on particle size (A: nanoparticle, B: microparticle, C: mixture of microparticles and nanoparticles).

## 4. Discussion

The medical radiation shield is mostly used as a material for radiation-shielding aprons, and the thickness and weight of the shielding fabric are important factors to ensure user safety. This study compared shielding performance by adjusting particle size to reduce the thickness and weight of the shielding sheet, and shielding performance was not efficient considering particle processing cost.

When a lightweight shielding suit for protection against low-dose exposure in medical contexts is manufactured, there is a sufficient protective effect even if it is not produced of nanosized particles. Radiation energy produces photoelectric and Compton effects during interactions within a shielded sheet [35].

The photoelectric effect is a phenomenon in which incidental photons interact with shielding particles and transfer the total energy of the photons to internal electrons, and internal electrons collide with surrounding atoms, causing scattering [36,37]. A large number of particles in the material increases the occurrence of scattering, and absorption in the shielding sheet accordingly increases because of the photoelectric effect in the low-

energy region; thus, good shielding efficiency is achieved. However, in the high-energy region, shielding is less effective because of scattering due to transmission rather than due to absorption. To improve shielding performance, more particles are required to increase the probability of interaction within the same area. Therefore, the shielding efficiency of the nanoparticle sheet has a temporary synergistic effect at high-energy intensity, as shown in Figure 6. That is, because the number of particles per unit area is large in the shielding sheet, density is high, and uniform dispersion is achieved [38]. By reducing DBP, there was an increase in the probability of interaction with X-ray photons in the particles. This result was caused by the clustering effect, and it can be assumed that the base material directly affects the particle distribution of the shielding material.

In particular, when mixing with a polymer series, the clustering effect can be predicted to be high, as the particle size is small. Future studies should compare the effects with other materials, such as rubber [39]. Furthermore, prior studies showed that the mixing of particle size and the base material that disperses it is crucial in the shielding-sheet manufacturing process, and that small particles generally exhibit high dispersion forces [40].

In this study, the change in shielding efficiency according to particle size and particle-size mixing was quantitatively analyzed through an experiment using tungsten particles, an ecofriendly material that could replace lead. In addition, in the case of producing lightweight shielding suits used by medical institutions, experiments demonstrated which particle composition of shielding materials is advantageous according to the energy domain. Consequently, when the shielding sheet was manufactured under the same conditions, the change in shielding efficiency relative to the particle size was not large; however, the smaller the particle size was, the more effective the high-energy shielding. Therefore, when manufacturing a shielding body used for resisting high-energy radiation, such as gamma rays, the particle size of the shielding material should be small. In addition, the mixability and combination of the base material that could narrow the gap between particles are important.

## 5. Conclusions

In the manufacture of medical radiation shields, there is not a large difference in shielding efficiency according to the particle size (between nanoparticles and microparticles); thus, the economic feasibility of commercialization is insufficient. When the two types of particles were mixed, shielding efficiency was not excellent, and it was the same as when a single particle-size type was used in the manufacturing process. In addition, nanoparticle shielding sheets resulted in a 5% increase in shielding efficiency in high-energy regions, and the difference between microparticles and nanoparticles was almost equivalent in the production of low-energy shielding.

**Funding:** This work was supported by the Radiation Technology R&D program through the National Research Foundation of Korea funded by the Ministry of Science and ICT (2020M2C8A1056950).

**Institutional Review Board Statement:** Not applicable.

**Informed Consent Statement:** Not applicable.

**Data Availability Statement:** Data are contained within the article.

**Conflicts of Interest:** The author declares no conflict of interest.

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
