# Peer review of "Analysis of Shielding Performance of Radiation-Shielding Materials According to Particle Size and Clustering Effects"

_applsci, doi:10.3390/app11094010_

Round 1

Reviewer 1 Report

The manuscript applsci-1152955 has been reviewed. It presents original scientific work on the analysis of medical radiation shielding performance in a non-uniform radiation shielding material with different particle sizes.

General comments:

The manuscript is generally well written even though it needs some revision by a native English speaking person. I give a non-exhaustive list of possible changes in the text.

The terms monodispersive and polydispersive are deprecated by the IUPAC, which recommends the use of uniform and non-uniform instead. (see Pure Appl. Chem. 81 (2): 351–353. DOI:10.1351/PAC-REC-08-05-02).

Abstract

The sentence lines 19 to 20 sounds a little bit confusing. What do you mean with “when a shielding sheet using a polymer substance is produced in the high-energy region”?

Introduction

  1. 2, line 52: please check the term “mold molding”.

Materials and Methods

  1. 2, lines 83-84: What do you mean with “The size and arrangement of particles can react actively with radiation; therefore, technology is required to increase the shield durability.”? Does the shield deteriorate with progressing use of it? How would this happen? Radiation damage?
  2. 3, line 91: What do you mean with the term “uniformly isotropic shielding material particles”?
  3. 3, lines 91-122: The various equations are not explained properly in the text, thus they are not easily understandable by the reader.
  4. 3, line 128: Of course these values are valid for the bulk material.
  5. 3-4, lines 129-131: There is something missing in the sentence. As it is now, it does not make much sense. Did you produce the micro- and nanoparticles by yourself?
  6. 4, table 1: Are you sure about the data given in this table? How did you calculate the specific surface area?
  7. 4, line 140: the molecular weight should be given in g/mol. Be more precise than “four million or more”.
  8. 4, line 143: change “as the solvent” to “as solvent”.
  9. 5, lines 167-169: it would be better to give more details on the experimental procedure, i.e. what field size was used at the level of the sheets, how where the X-rays characterized (which filters were used, their energies). What do you mean when you write that the dosimeter was “used after the inspection and calibration”? Why did you specifically chose the distances reported in Fig. 3? In Fig. 3 correct “Shieding” to ”Shielding”.
  10. 5, throughout the remaining text: instead of “shielding rate” better use “shielding efficiency”.
  11. 5, lines 183-185: this sentence is misleading, you do not observe changes in particle size etc. you observe changes in the shielding efficiency due to the use of shielding sheets with different properties (particle size, array composition).

By the way, which different array compositions did you use?

Results

  1. 5, line 188: the text is somewhat misleading you could reformulate it to “the composition as that reported”; what do you mean with “composition ratio”?
  2. 56, Table 2: In the table you cite polyurethane, maybe you meant polyethylene.
  3. 6, lines 198-201: The sentence is not clear. Do you mean that the structure was not homogeneous?
  4. 6, Fig. 4: if the first picture in each row should show the structure close to the surface, for sample (C) this is not shown. At least to the eye there is no clear difference between samples (A), (B) and (C). It would be indicated the use of a software for image analysis in order to distinguish quantitatively the three different structures and their clustering.
  5. 7, Fig. 5: At least to the eye there is no clear difference between samples (A), and (B). Also here would be indicated the use of a software for image analysis in order to distinguish quantitatively the two different structures and their clustering. The text referring to this figure on p. 5, lines 204-215 is rather qualitative and need some quantitative confirmation.
  6. 7, Fig. 6: it would be good to have error bars in this figure, or at least, if they are too small, a statement about it in the figure caption. The difference of 5 % is significant with respect to the uncertainties?

Discussion

  1. 7, lines 233-235: Where did you discuss this point in the manuscript? No mentioning was done until now about the particle processing cost.
  2. 7, lines 238-239: This is a very general statement. Why do you make it here?
  3. 7, line 242: causing scattering of what?
  4. 7, line 244: “energy of a low region” = “low energy region”?
  5. 7, line 245: “energy in a high region” = “in the high energy region”? How do you distinguish exactly high and low energies?
  6. 8, line 246: what do you mean exactly with the term transmission?
  7. 8, line 252: Where do you get the free electrons from?
  8. 8, lines 252-254: This statement is not clear at all. Isn’t clustering the opposite of uniform dispersion? What do you mean with the “role of the base material”? How does this “role” affect the particle distribution of the shielding material?
  9. 8, lines 263-265: This statement is not clear. Please rephrase it.
  10. 8, line 267: Maybe replace “and” with “but”. Could express better what you saw.
  11. 8, line 269: Why do you refer now to gamma rays? You used an X-ray tube.

Conclusions

  1. 8, line 275: It is not clear wherefrom you derive the statement about feasibility and commercialization.
  2. 8, line 276: You say here the shielding efficiency was not excellent, with respect to what? In the manuscript you did not compare it to some standard you want to reach (e.g. equivalent lead shield or similar).

Final comments

This manuscript is presenting original and interesting research. Unfortunately it suffers a little bit of confusion and does not go enough into detail of the experiment. Moreover, some qualitative statements are rather weak and should rather be made quantitatively to give the readers more information and to make them follow better the line of thought of the author. For the sake of getting the most out of this work a major revision is recommended. The authors are invited to go through the comments, as the manuscript by itself is interesting and worth publishing it.

Reviewer 2 Report

The article demonstrates interesting results regarding the effect of particle size on the shielding performance for medical application. There are some problems that need to be resolved in order to be considered for publication.
1.    It’s better to shorten the title.
2.    Page 2 Line 81: what does collision of particles mean? Since radiation interact with electronic field, I’m not sure what the particles here?
3.    Page 3 Line 111: \mu x is not the mean free path. There is some fundamental misunderstanding here.
4.    Equation 5: Please define \mu_i x_i here?
5.    Table 1 particle size distribution for microparticle and nanoparticle is not consistent with Figure 4 and Figure 5. In the figures, the nanoparticle and microparticle are almost on the same order of size.
6.    Page 5 Line 190: “a larger amount of shielding material can be added to a sheet composed of only nanoparticles.”, then why the sheet weight per unit area is lower in Table 2 for A.
7.    Page 7 Line 220: “the larger the particle size, the greater the shielding rate”, this is contradictory with “if the size of the particles is small, …,  the shielding rate can be expected to improve.” (Line 212-213) and abstract.
8.    Figure 6: no error bar?
9.    Page 7 Line 242: how electrons collide with surrounding atoms?
10.    Page 8 Line 279: low-dose shielding or low-energy shielding?

Reviewer 3 Report

In this study, to observe how the particle size of the shielding material affects the shielding performance, performance and particle structure were observed for each shielding sheet made by mixing microparticles and nanoparticles with a polymer material using the same process.

The paper’s subject is very practical and interesting. The research procedure has been logically carried out and includes experimental work. Therefore, I highly recommend this paper for publication in this journal but before that, I have some few comments on the text:

Comments:

1)The texts in figure 1 have overlap with the plot. I recommend to put the text besides the figure rather than on them.

2)Caption of figure is incomplete. Please explain figures a,b, and c in the caption.

3)Please explain in the manuscript the used parameters (x-ray tube voltage, current, time of exposure, etc.) for exposure of shield samples shown in figure 3.  

4)Did you consider the photon scattering effects in your experiments? The scattered photons from around medium (which are not related to sample sheet) can have effect on the measured dose in the dosimeter.

5) Recently, artificial intelligence has been widely used for optimizing the radiation based systems and has some advantages over traditional methods. I recommend the authors to add some references in the paper in field of application of artificial intelligence for optimizing the photon radiation based systems both in field of medical and industry. Some suitable and new references are listed in the following:

[1] McCollough, C.H. and Leng, S., 2020. Use of artificial intelligence in computed tomography dose optimisation. Annals of the ICRP, 49(1_suppl), pp.113-125.

[3]Takam, C.A., Samba, O., Kouanou, A.T. and Tchiotsop, D., 2020. Spark Architecture for deep learning-based dose optimization in medical imaging. Informatics in Medicine Unlocked, p.100335.

[4] Roshani, M., Phan, G.T., Ali, P.J.M., Roshani, G.H., Hanus, R., Duong, 2021. Evaluation of flow pattern recognition and void fraction measurement in two phase flow independent of oil pipeline’s scale layer thickness. Alexandria Engineering Journal.

[5]Roshani, M., etc., 2021. Combination of X-ray tube and GMDH neural network as a nondestructive and potential technique for measuring characteristics of gas-oil–water three phase flows. Measurement, 168, p.108427.

Round 2

Reviewer 2 Report

The following questions need to be resolved for publication:
1.    Page 3 Line 129: 400-900 nm not consistent with the value in Table 1 for nanoparticle
2.    Page 3 Equation 5: provide definition for mu_i and x_i
3.    Page 3 line 111: the definition of mean free path is wrong. I suggest the author reading some wiki: https://en.wikipedia.org/wiki/Mean_free_path#Radiography

Reviewer 3 Report

Although most the required comments were addressed correctly, some comments were responded/added in the text incompletely. I highly recommend the author to do these comments:

1) The question about considering back scattering effects was responded correctly, but it should be added in the text. It should be clear for readers. So please put whole/part of your following explanation in the text:

“The radiation shielding test conducted in this study was evaluated by applying the lead equivalent test method (KS A 4025: confirmed in 1990, 1995) of X-ray protection products of the Korean industrial standard. In addition, the photon scattering effect was measured at a distance of 50 cm, as shown in Fig. 3, to prevent backscattering on the detector.”

In page 1, line 29, only references related to application of artificial intelligence in medical radiation are mentioned. I highly recommend to add also some new references regarding application of AI in industrial radiation. Some good references are recommended in the following. Also you can rewrite the sentences in lines 28-29 in this following way:

“As the field of application of medical and industrial radiation is expanding, the radiation exposure of medical [1,2] and industrial [3,4] diagnostic devices under artificial radiation is also increasing”

[1] and [2]: same as mentioned in manuscript.

[3] Roshani, M., Phan, G.T., Ali, P.J.M., Roshani, G.H., Hanus, R., Duong, 2021. Evaluation of flow pattern recognition and void fraction measurement in two phase flow independent of oil pipeline’s scale layer thickness. Alexandria Engineering Journal.

[4] Roshani, M., Phan, G., Faraj, R.H., Phan, N.H., Roshani, G.H., Nazemi, B., Corniani, E., 2020. Proposing a gamma radiation based intelligent system for simultaneous analyzing and detecting type and amount of petroleum by-products. Nuclear Engineering and Technology.

Round 3

Reviewer 3 Report

All the comments have been addressed correctly and paper is ready for publication in the present form!